# Sigma-2 Receptor—A Potential Target for Cancer/Alzheimer’s Disease Treatment via Its Regulation of Cholesterol Homeostasis

**DOI:** 10.3390/molecules25225439

**Published:** 2020-11-20

**Authors:** Kai Yang, Cheng Zeng, Changcai Wang, Meng Sun, Dan Yin, Taolei Sun

**Affiliations:** 1School of Chemistry, Chemical Engineering and Life Science, Wuhan University of Technology, 122 Luoshi Road, Wuhan 430070, China; zengchen0077@163.com (C.Z.); wangchangcai@whut.edu.cn (C.W.); 15556991710@163.com (M.S.); 2State Key Laboratory of Biocatalysis and Enzyme Engineering, School of Life Science, Hubei University, Wuhan 430062, China; 3Hubei Province Key Laboratory of Biotechnology of Chinese Traditional Medicine, National & Local Joint Engineering Research Center of High-Throughput Drug Screening Technology, Hubei University, Wuhan 430062, China; 4State Key Laboratory of Advanced Technology for Materials Synthesis and Processing, Wuhan University of Technology, 122 Luoshi Road, Wuhan 430070, China

**Keywords:** sigma-2 receptor, TMEM97, cholesterol homeostasis, Alzheimer’s disease, cancer

## Abstract

The sigma receptors were classified into sigma-1 and sigma-2 receptor based on their different pharmacological profiles. In the past two decades, our understanding of the biological and pharmacological properties of the sigma-1 receptor is increasing; however, little is known about the sigma-2 receptor. Recently, the molecular identity of the sigma-2 receptor has been identified as TMEM97. Although more and more evidence has showed that sigma-2 ligands have the ability to treat cancer and Alzheimer’s disease (AD), the mechanisms connecting these two diseases are unknown. Data obtained over the past few years from human and animal models indicate that cholesterol homeostasis is altered in AD and cancer, underscoring the importance of cholesterol homeostasis in AD and cancer. In this review, based on accumulated evidence, we proposed that the beneficial roles of sigma-2 ligands in cancer and AD might be mediated by their regulation of cholesterol homeostasis.

## 1. Introduction

The sigma receptors were identified in 1976 and initially recognized as a member of the opiate receptors; later studies revealed that they were different receptors [1]. The sigma receptors were classified into sigma-1 and sigma-2 receptors based on their different pharmacological profiles [2]. The sigma-1 receptor acts as a molecular chaperon and primarily localizes at the mitochondria-associated endoplasmic reticulum membrane (MAM) [3]. Upon stimulation by its ligands, the sigma-1 receptor translocates from MAM to plasma membrane (PM) [3], where it interacts with ion channels and G protein coupled receptors (GPCRs). The sigma-1 receptor plays important roles in many CNS diseases including drug addiction, depression, schizophrenia, Parkinson’s disease (PD), and Alzheimer’s disease (AD) [4,5,6,7]. The structure of the sigma-1 receptor has been recently reported, which exists as a trimer with one single transmembrane topology for each protomer [8]. In contrast, the molecular identity of the sigma-2 receptor was only recently revealed [9]. In 2006, Colabufo et al. proposed that sigma-2 receptors are histone proteins, since a specific sigma-2 ligand PB28 could pull down histone proteins [10]. However, this hypothesis was discarded. It showed that fluorescent ligands of the sigma-2 receptor are not found in the nucleus, which was not consistent with this histone hypothesis [11].

### 1.1. PGRMC1/Sigma-2 Receptor

In 2011, Xu et al. identified the sigma-2 receptor as a component of the progesterone receptor membrane component 1 (PGRMC1), because PGRMC1 could directly link to WC-21, which is a photoaffinity ligand for the sigma-2 receptor. PGRMC1 knockdown cells abolished this ligand binding to the sigma-2 receptor, while PGRMC1 overexpression increased the interaction between ligand and sigma-2 receptor [12].

However, later-accumulated evidence has suggested that the sigma-2 receptor is distinct from PGRMC1 [13]. The knockdown of PGRMC1 using siRNA or the overexpression of this protein using its cDNA in MCF7 cells did not alter the binding of [^3^H]-1,3 di-ortho-tolylguanidine ([^3^H]-DTG)to the sigma-2 receptor [14]. Knocking out PGRMC1 using CRISPR/Cas9 techniques in NSC34 cells also drew the same conclusion [15]. Furthermore, the molecular size of PGRMC1 is different from that of the sigma-2 receptor. Using photoaffinity labeling technique, the labeling with [^3^H] DTG in rat livers reveals that the sigma-2 receptor is about 21.5 kDa, while the molecular size of PGRMC1 is 25 kDa [16]. In addition, the sigma-2 receptor could still be detected by the photolabeling using [^125^I]-iodoazido-fenpropimorph ([^125^I]-IAF) in the PGRMC1 knockout cells [14]. Fluorescent sigma-2 ligands still have ability to bind to their receptor without the expression of PGRMC1 [17].

### 1.2. TMEM97/Sigma-2 Receptor

Recently transmembrane protein 97 (TMEM97) was identified as a sigma-2 receptor identity using mass spectrometry after affinity purification from the liver [9]. TMEM97 is an ER-resident transmembrane protein, also known as meningioma-associated protein (MAC30) [18]. The reduction in TMEM97 expression using siRNA decreased the binding of the sigma-2 receptor to its ligand [^3^H] DTG. In addition, the overexpression of TMEM97 in cells lacking the sigma-2 receptor demonstrated a similar sigma-2 receptor binding profile. The affinity of ligands for TMEM97 is identical to the sigma-2 receptor binding affinity [9]. Moreover, TMEM97 ligands bind sigma-2 receptors: Asp29 and Asp56 are identified as ligand binding sites [9].

TMEM97 is expressed in many cell types and play important functions in neurons and cancer cells [19,20,21,22]. It has been shown that TMEM97 is involved in alcohol withdrawal behaviors. JVW-1034 (a TMEM97 ligand) prevented withdrawal-induced behavioral impairments in worms and blunted withdrawal-induced excessive alcohol drinking in rats [21]. TMEM97 is also involved in neuropathic pain: the TMEM97 receptor agonist, UKH-1114, reduces mechanical hypersensitivity in an animal model of neuropathic pain [20]. CM398, a selective sigma-2 ligand, also showed anti-inflammatory analgesic effects in formalin model of inflammatory pain in mice [23]. Further, the activation of the sigma-2 receptor is neuroprotective: a novel sigma-2 receptor/TMEM97 modulator, DKR-1677, protects neurons from death and cognitive impairment after blast-mediated traumatic brain injury (TBI) [22]. Additionally several compounds with sigma-1/sigma-2 mixed selectivity were able to counteract the neurotoxicity induced by oxidative stress [24]. These evidences indicate that the sigma-2 receptor/TMEM97 may be a promising target for the treatment of neurological diseases. TMEM97 is also involved in cancer: its agonist, PB221, significantly inhibited the migration and invasion of ALTS1C1 cells (a murine brain tumor cell line). In addition, a high dose of PB221 induced cell death of ALTS1C1 cells; these effects all involved mitochondrial oxidative stress. Consistent with this notion, PB221 effectively retarded tumor growth in tumor models [19]. Using CRISPR/Cas9 technology to remove TMEM97 in HeLa cells, one recent study has found that TMEM97 does not mediate cytotoxicity induced by the sigma-2 ligand [25]. It is possible that this cytotoxicity might be caused by sigma-2-ligand-induced lysosomal dysfunction and reactive oxygen species (ROS) production, but whether it relies on sigma-2 receptor remains unstudied.

### 1.3. The Complex of Sigma-2 Receptors

In fact, TMEM97/sigma-2 receptor forms a trimeric complex with PGRMC1 and low-density lipoprotein (LDL) receptor (LDLR), which is responsible for the efficient uptake of LDL into the cells [26]. This complex is also necessary for the internalization of apolipoprotein E (ApoE) and Aβ monomers and oligomers [27]. These findings explain why PGRMC1 could be still photoaffinity tagged with a sigma-2 ligand, although PGRMC1 is not the sigma-2 receptor [12].

Consistently the activation of TMEM97 by its ligands could affect PGRMC1-dependent cell processes [21,28]. For example, JVW-1034 reduces withdrawal-induced impairments in ethanol-treated worms via PGRMC1/VEM-1 (a PGRMC1 ortholog in worms), although whether JVW-1034 binds directly to PGRMC1 is not studied [21]. The Sigma-2/TMEM97 ligand, SAS-0132, modulates PGRMC1-dependent mechanisms to reduce cell death, cognitive deficits, and neuroinflammation in AD mice [28].

### 1.4. The Pharmacology of Sigma-2 Receptors

To date, numerous sigma-2 selective ligands have been developed and characterized by receptor binding assays (Table 1). The definition of a sigma-2/TMEM97 agonist and antagonist remains undefined. Zeng et al. proposed that sigma-2 selective compounds with cytotoxic effects on cancer cells are categorized as agonists, since siramesine, a commonly accepted sigma-2 agonist, induces cytotoxicity in cancer cells [29]. Using this approach, sigma-2 ligands have been divided into agonists, partial agonists, and antagonists [29]. However, molecular basis for pharmacological mechanism of action of sigma-2 ligands is not understood.

## 2. Sigma-2 Receptor—A Novel Regulator of Cholesterol Homeostasis

### 2.1. Cholesterol Synthesis

Cholesterol is required for not only membrane integrity and fluidity but also the production of hormones including steroids and vitamins [45,46]. Cholesterol biosynthesis begins with acetyl-CoA in the cytoplasm. Acetoacetyl-CoA is produced from two acetyl-CoA using acetyl-CoA acyltransferase 2 and then reacts with the third acetyl-CoA by the HMG-CoA synthase to produce 3-hydroxy-3-methylglutaryl-CoA (HMG-CoA). HMG-CoA reductase (HMGCR) reduces HMG-CoA to mevalonate, which is the primary rate-limiting step in cholesterol synthesis [45,46]. Mevalonate is converted to farnesyl pyrophosphate (FPP) by a series of enzymatic reactions, then two FPPs are condensed to squalene, which commits to sterol production. Most of squalene is converted to lanosterol, and through several additional enzymatic reactions, lanosterol is converted to cholesterol. FPP is also converted to geranylgeranyl pyrophosphate (GGPP); both FPP and GGPP modify oncogenic proteins, such as Ras, via enzymatic prenylation and activate them [45,46] (Figure 1).

The synthesis of cholesterol is regulated by feedback control involving sterol regulatory element-binding protein 2 (SREBP2). SREBP is an ER membrane-bound protein, where it interacts with SREBP cleavage-activating protein (SCAP) [47]. When the sterol level at the ER is low, the conformation of SCAP is changed, allowing SREBP2 to translocate from the ER to the Golgi. At the Golgi, SREBP2 undergoes two sequential cleavages by site 1 protease (S1P) and site 2 protease (S2P), liberating the active soluble fragment of SREBP2 from the membrane. The processed SREBP2 enters the nucleus and increases expressions of many genes involving sterol biosynthetic pathway including HMGCR. Conversely the accumulation of cholesterol in the ER inactivates the SREBP2 pathway via insulin-induced genes (INSIGs). INSIGs interact with SCAP and promote the ER retention of SCAP, which blocks the ER-to-Golgi transportation of SREBP2 and reduces cholesterol biosynthesis [47] (Figure 2). In addition to SREBP2, liver X receptors (LXRs) [48,49] and nuclear factor erythroid 2 related factor-1 (NRF1) [50] are also involved in the regulation of cholesterol synthesis. High cholesterol levels activate LXRs, resulting in cholesterol synthesis inhibition [48,49].

### 2.2. Cholesterol Transport

In addition to de novo cholesterol synthesis, cells can acquire cholesterol from an external source. Cholesterol can be obtained from LDL, which is the major cholesterol-carrier in the blood via the clathrin-mediated endocytosis of LDLR. In the brain, neurons acquire their cholesterol mainly from apolipoprotein E (ApoE), the major lipoprotein in the CNS, via multiple ApoE receptors including LDLR, LDLR-related protein 1 (LRP1), and the very-low-density lipoprotein receptor (VLDLR) [51]. Once in the late endosomes/lysosomes in the cytosol, LDL is released from LDLR, then LDLR is recycled back to the cell surface, while LDL is degraded in the lysosome, and cholesterol is released. When the cell’s cholesterol demand is no longer high, proprotein convertase subtilisin/kexin type 9 (PCSK9) is secreted, which directs LDLR to the lysosome for degradation [52,53]. After cholesterol is released, it is then transported to the destined membranes to meet the need of cells [54]. When intracellular cholesterol is in excess, it is either exported out of cells by ATP-binding cassette (ABC) transporters including ABCA1 and ABCG1 or becomes esterified to form cholesteryl esters (CE) by acyl-coenzyme A: cholesterol acyltransferases (ACATs) [45]. Extracellularly high-density lipoproteins (HDL) remove excess cholesterol from tissues to the liver through reverse cholesterol transport (RCT), where it is subsequently eliminated with bile [55].

Cholesterol can also be oxidized via enzymatic or non-enzymatic reactions to produce oxysterols [56]. Cholesterol yields 7-ketocholesterol (7KC) and 7β-hydroxycholesterol (7βHC) by autoxidation, which does not need the help of enzymes. Other oxysterols are produced enzymatically by members of the cytochrome P450 family; for example, 25-hydroxycholesterol (25-HC) is produced by cholesterol 25-hydroxylase (CH25H), 24-hydroxycholesterol (24-HC) by CYP46A1, and 27-hydroxycholesterol (27-HC) by CYP27A1, respectively [57]. These oxysterols could act as ligands to activate LXRs, which regulate lipid metabolism and efflux [58]. In addition, oxysterols also have roles in immunity and inflammation. In particular, oxysterols, such as 25-HC, have been shown to have both pro- and anti-inflammatory effects on immune response. On the one hand, it can act as a signaling molecule to amplify inflammatory activation in macrophages [59]: 25-HC increases the production of many pro-inflammatory cytokines and chemokine including IL-6 and IL-8 [60]. On the other hand, the production of 25-HC can prevent absent in melanoma 2 (AIM2) inflammasome activation in macrophages [61]. In conclusion, the roles of 25-HC in inflammation is quite complex, and further studies are strongly required.

Lipoprotein LDL can also be modified by oxidation. This modified LDL (oxLDL) is a ligand for Toll-like receptors (TLRs) in macrophages, which directly activates pro-inflammatory signaling pathways [62]. Further, macrophages also engulf oxLDL and cause the accumulation of cellular cholesterol in macrophages, resulting in the amplification of TLR signaling [63,64].

### 2.3. The Involvement of the Sigma-2 Receptor in Cholesterol Homeostasis

Substantial evidence has shown that the sigma-2 receptor/TMEM97 is involved in the synthesis of cholesterol. It was reported that, after treatment with progesterone, TMEM97 and cholesterol biosynthesis genes are coordinately upregulated in normal ovarian surface epithelial cells [65]. TMEM97 shares EXPERA functional domain with other cholesterol-related genes including transmembrane 6 superfamily member 2 (TM6SF) and emopamil binding protein (EBP) [18], implicating that TMEM97 is a cholesterol synthesis gene. Consistently, using an RNAi screening technique, TMEM97 was identified as a regulator of cholesterol homeostasis. The knockdown of TMEM97 reduced cholesterol contents as well as the internalization of LDLR. Furthermore, under sterol-depleted conditions, TMEM97 mRNA is upregulated, indicating TMEM97 plays a very important role in the regulation of cholesterol homeostasis [66].

TMEM97 also controls the trafficking of cholesterol. TMEM97 could form a trimeric complex with PGRMC1 and LDLR and mediate cholesterol uptake via its interaction with PGRMC1 and LDLR: the loss or inhibition of either one of these proteins results in a decreased uptake of cholesterol [26]. In addition, TMEM97 is a Niemann-Pick C1 (NPC1) binding protein, which is required for transporting cholesterol out of lysosomes [66]. The loss of this protein results in NPC, a fatal lysosomal storage disorder. In NPC cellular model, TMEM97 knockdown upregulates NPC1 expression, reduces cholesterol accumulation, and restores the trafficking of cholesterol out of the lysosome [67] (Figure 3).

## 3. Anti-Cancer Effects of Sigma-2 Receptor Ligands

### 3.1. Increased Cholesterol Synthesis and Uptake

Cellular cholesterol is usually acquired from both synthetic pathways and diet; most cancer cells exhibit increased cholesterol synthesis and uptake [46]. In cancer, multiple enzymes involved in mevalonate pathway are upregulated, including HMGCR and SREBP, which produce more cholesterol for cell proliferation [68]. Cancers are characterized by a gain of oncogenes and loss of tumor suppressors. P53, a tumor suppressor, could block SREBP activation and reduce cholesterol synthesis [69]. In addition to de novo cholesterol biosynthesis, some cancer cells increase cholesterol uptake, which is more efficient and requires less ATP consumption. Usually for these cells, they have a high expression of LDLR [70].

### 3.2. Enriched Cholesterol-Derived Metabolites: Oxysterols

In the presence of excessive cholesterol, a high amount of oxysterol is produced in cancer cells. However, the exact effect of oxysterols on carcinogenesis and cancer progression is quite complicated. Many studies have showed that oxysterols can play precancerous and pro-proliferative roles in cancer cells. In patients with estrogen-receptor-positive breast cancer, oxysterol 27-HC is elevated [71]. In addition, the application of 27-HC reduces the expression of E-cadherin and β-catenin in breast carcinoma MCF7 cells, implicating that 27-HC is involved in the epithelial–mesenchymal transition (EMT) [72]. In addition, 27-HC promotes cell proliferation in MCF7 cells [73]. These effects can be explained by the modulation of signaling pathways by oxysterols; 27-HC could reduce p53 activation by enhancing the function of p53 E3 ligase, murine double minute 2 (MDM2), and promoting cell proliferation [73]. It could activate signal transducer and activator of transcription 3 (STAT3)-vascular endothelial growth factor (VEGF) signaling and facilitate angiogenesis [74].

Oxysterols also have anti-cancer effects and induce the death of tumor cells. The pro-apoptotic effect of oxysterols is caused by the overproduction of ROS and/or the increase in Ca^2+^ level in the cells [75]. Indeed, 27-HC could inhibit gastric cancer cell proliferation and migration via the modulation of LXR signaling [76]. Similarly, 27-HC treatment impedes cell proliferation in colorectal cancer cells, but this effect is mediated by the dephosphorylation of the kinase Akt [77].

### 3.3. The Involvement of Sigma-2 Receptor in Cancer

#### 3.3.1. Sigma-2 Receptor as an Imaging Target for Cancer Diagnosis

The sigma-2 receptor is expressed in higher density in proliferating tumor cells compared to quiescent tumor cells. Using the mouse mammary tumor 66 cell line, the density of sigma-2 receptors was found to be 10-fold higher in proliferating 66 versus quiescent 66 cells in vitro [78], later this conclusion is confirmed in solid tumor xenografts [79], suggesting that the sigma-2 receptor might be a promising marker for the proliferative solid tumors. Up to now, both ^11^C- and ^18^F-radiolabeled sigma-2 receptor ligands have been developed and validated in a variety of tumor models [80,81]. [^18^F]ISO-1 has been tested in humans for tumor positron emission tomography (PET) imaging [80,82] and used as a predictor of the cancer therapy response [83]. In 28 breast cancer patients, the uptake of [^18^F] ISO-1 was significantly correlated with expression of tumor proliferation marker Ki-67 [82].

#### 3.3.2. Sigma-2 Ligands as Anticancer Agents

Sigma-2 ligands can inhibit cancer cell proliferation, suppress tumor growth, and induce tumor cell death [19,84]. Many studies have shown that they are cytotoxicity in cancer cells. For example, using propidium iodide (PI) staining technique, it has been shown that NO1, a fluorescent sigma-2 receptor ligand, enhances apoptosis in breast cancer cell lines. Further in cells with overexpression of TMEM97, more death of cancer cells can be induced by NO1. In addition, NO1 reduces cell proliferation and migration in triple-negative breast cancer cells [84]. It is proposed that NO1 can downregulate stromal interaction molecule (STIM1)-Orai1 interaction and reduce store-operated calcium entry (SOCE) in cancer cells [84]. Additionally, PB221, a sigma-2/TMEM97 receptor ligand, significantly enhances cell death and has antiproliferation activity against brain tumor cells [19]. This effect is mediated by mitochondrial oxidative stress. It also retards the migration and invasion of invasive murine astrocytoma cells in vitro. Furthermore, in vivo study reveals that PB221 effectively inhibits tumor growth [19]. However, in these studies, if the sigma-2 receptor is involved, it is not studied.

Recently, Zeng et al. reported that neither sigma2/TMEM97 nor PGRMC1 mediates sigma2-ligand-induced cytotoxicity. Instead, sigma-2-ligand-induced lysosomal dysfunction and ROS production may be responsible for sigma-2-ligand-induced cytotoxicity [25]. However, whether TMEM97 or PGRMC1 mediates sigma-2-ligand-induced lysosome dysfunction and ROS production needs to be studied. Considering a large difference in the affinity of ligands for the sigma-2 receptor (nM) and their efficacy in cytotoxicity assays (µM), the anti-cancer mechanism of action of putative sigma2 selective compounds remains unclear.

#### 3.3.3. Sigma-2 Receptor Ligands as Anticancer Drug Delivery Vehicles

In order to reduce side effects caused by non-targeted chemotherapeutic agents, the sigma-2 ligand has also been used as a vesicle to deliver anticancer drug to cancer cells precisely [85,86,87], it could deliver small molecules by its internalization into cancer cells.

Two different approaches using sigma-2 ligand-based drug delivery have been developed. First, sigma-2 ligands are conjugated with various nanoparticles, which are filled with cytotoxic agents. Telmisartan (TEL) is a cytotoxic agent that could inhibit the prostate cancer by the augmentation of apoptosis [88], but it has several dose-dependent side-effects including renal dysfunction and myocardial infarction [89], which hamper its wide acceptance. In order to improve the targeting of TEL, sigma-2 receptor ligand, 3-(4-cyclohexylpiperazine-1-yl) propyl amine (CPPA), was linked to nanostructured lipid particles containing TEL (CPPA-TEL-NLPs) (Figure 4A), CPPA-TEL-NLPs enter the PC-3 cells via sigma-2-receptor-mediated endocytosis and subsequently activate multiple apoptosis pathways to kill cancer cells. This construct demonstrated superior cytotoxicity and great cellular uptake in PC-3 cells [87].

Second, sigma-2 ligands were covalently linked to antisense oligonucleotides or antitumor peptides. Erastin is a small molecule capable of inducing ferroptosis, it inhibits the cystine-glutamate antiporter system Xc- and prevents cells from synthesizing the antioxidant glutathione, which results in excessive lipid peroxidation and cell death [90]. Although Erastin and its analogues have cancer-selective cytotoxic activity, they lack effectiveness for pancreatic cancer patients; this might be caused by a deficiency in cellular drug uptake. In order to solve this issue, the Erastin derivative des-methyl Erastin (dm-Erastin) was chemically linked to the sigma-2 ligand SV119 to create SW V-49 (Figure 4B). This conjugation increases the killing capacity of dm-Erastin in vitro. Further, SW V-49 overcomes the cellular internalization block of dm-Erastin and reduces tumor sizes [86].

In addition, SW IV-134 is constructed by the conjugation of a sigma-2 receptor ligand (SW43) and small molecule second mitochondria-derived activator of caspases (SMAC) mimetic compound (SMC) (Figure 4C). SMC is also called inhibitor of the apoptosis protein (IAP) antagonist; it has the ability to suppress IAPs and reestablish the apoptotic pathways [91]. SW IV-134 is tested in triple-negative breast cancer. It showed SW IV-134 can induce cytotoxicity that exceeds the most commonly used drug in breast cancer therapy [85].

### 3.4. Mechanisms of Anti-Cancer by Sigma-2 Ligands Targeting Cholesterol Homeostasis

It has been proposed that the anti-cancer effects of sigma-2 ligands might depend both on the sigma-2 ligand used and on the cell type, they involve caspase-dependent and -independent apoptosis, Ca^2+^ overload, ROS generation, lysosomal membrane permeabilization (LMP), and autophagy [92,93,94,95]. In fact, more and more recent evidence provides another possibility that sigma-2 ligands might target cholesterol homeostasis to treat cancer [96].

In cancer cells, accumulated cholesterol could form more lipid rafts and activate various cellular signaling pathways, which promote cancer development [97,98]. Lipid rafts are highly ordered membrane domains consisting of cholesterol and sphingolipids. They have the ability to modulate membrane fluidity, lateral movement of proteins as well as signal transduction [99]. The compartmentalization of signaling pathway enhances the efficiency of signal transduction. For example, cholesterol activates the sonic hedgehog (SHH) pathway and promotes cell cycle progression, which contributes to cancer development [98]. Cholesterol also increases transforming growth factor β (TGF-β) signaling. TGF-β is required for the induction of EMT in cancer cells. Further, TGF-β receptor in lipid raft activates the mitogen-activated protein kinase (MAPK) pathway, which favors cancer cell proliferation and migration [97]. Sigma-2 ligands could disorganize the lipid rafts in the membrane by displacing cholesterol molecules and dampen the lipid raft microdomain-mediated signaling. Consistent with this notion, the depletion of cholesterol from these lipid rafts enhances apoptotic death of cancer cells [100].

Furthermore, the tumor microenvironment is very important for cancer development. Cholesterol metabolites, oxysterols, affect immune cells in the tumor microenvironment. Oxysterols inhibit T cell anti-tumor ability via LXR activation [101]. Furthermore, oxysterols promote tumor metastasis. In addition, 25-HC interacts with EBI2 (a GPCR that directs the migration of immune cells in response to oxysterols) and triggers migration of both macrophages and monocytes [102]. In a breast cancer model, 27-HC has been found to attract polymorphonuclear neurotrophils and γδ T cells [103]. Sigma-2 ligands might play an anti-cancer role by reducing the production of oxysterols and inhibit cancer development.

## 4. Sigma-2 Receptor Ligands May Target Cholesterol Homeostasis to Treat AD

AD is the most common neurodegenerative disease in the elderly [104]. It is characterized by extracellular accumulation of Aβ and intracellular deposits of hyper-phosphorylated tau protein [104]. Aβ is produced via a two-step cleavage of the amyloid precursor peptide (APP) by β secretase and γ secretase (also called presenilin (PS)) [105]. Abnormal lipid metabolism has been observed in AD [106]. Many genes associated with the regulation of lipid metabolism are also linked to the risk of developing sporadic AD, including clusterin (CLU), ATP-binding cassette subfamily A member 7 (ABCA7), sortilin-related receptor 1 (SORL1), and triggering receptor expressed on myeloid cells 2 (TREM2) [107]. In addition, the ε4 allele of ApoE is identified as the most important risk gene for late-onset sporadic AD [108].

### 4.1. Cellular Cholesterol Accumulation in AD

The blood–brain barrier (BBB) consists of tight junctions between the endothelial cells of brain and blood vessels, which prevents cholesterol uptake from the periphery, so brain cholesterol is synthesized in situ in the brain. Although all types of brain cells can synthesize cholesterol during development, but neurons in the adult do not efficiently synthesize cholesterol, they rely on the input from astrocytes as an external source [109].

In AD, cholesterol synthesis is reduced. Aβ fibrils reduce cholesterol synthesis in cultured neurons [110]. In addition, Aβ42 prevented the cleavage of SREBP-2 by protease and inhibited the transcription of many proteins including HMGCA, which are required for cholesterol synthesis [111]. Surprisingly, despite the reduced cholesterol synthesis, cellular cholesterol content is increased in AD [112]. It might be caused by the reduction in cholesterol conversion to 24-HC or increased cholesterol uptake. High levels of cholesterol have been suggested as a risk factor for AD [113]. For patients taking statins, which are cholesterol lowering drugs, the prevalence of AD is reduced [114,115], but other studies have obtained the opposite conclusion [116].

### 4.2. Altered Level of Oxysterols in AD

To date, the oxysterols including 24-HC and 27-HC are implicated in the pathogenesis of AD. In neurons, a brain-specific enzyme CYP46A1 converts excessive cholesterol into 24-HC, then it is exported out of the brain and carried by LDL to the liver for degradation, while another oxysterol, 27-HC, is produced by CYP27A1 in the periphery and moves into the brain by circulation. Then, 27-HC is converted into 7α-hydroxy-3-oxo-4-cholestenoic acid (7−OH-4-C) by the enzyme CYP7B, which then diffuses out of the brain through the BBB and moves into the periphery for degradation [109] (Figure 5).

Similar to cholesterol, the levels of oxysterols are also altered in AD. It has been shown that 24-HC can both increase [117] and decrease [118] in the brain. It is proposed that 24-HC is elevated in early AD but decreases later when neurons that express CYP46A1 die. In addition, 27-HC is elevated in AD brains [119]: accumulation of 27-HC in the brain is due to the increased flux of this oxysterol from the periphery across the BBB. Further, 27-HC could be synthesized in situ, because CYP27A1 is also expressed in astrocytes and oligodendrocytes [120]. Additionally, because neurons expressing CYP7B die, reduced degradation of 27-HC also contributes to its accumulation [121].

### 4.3. The Involvement of Sigma-2 Receptor in AD

#### 4.3.1. Sigma-2 Receptor Ligands as Therapeutic Target of AD

Using an in vitro quantitative receptor autoradiography technique, both sigma-1 and sigma-2 receptors are found to be widely distributed in the rat brain, but the expression levels of sigma-2 receptor are generally lower than those of sigma-1 receptor [122]. In addition, high amounts of sigma-2 receptors are located in the substantia nigra pars reticulata (SNr), cerebellum, and the motor cortex [122].

In AD, the expression of the sigma-2 receptor is altered. In female mice with a double Aβ deposition (APP/PS1), sigma-2 expression was significantly reduced compared to that in control [123]. However, another study found that there is no difference in sigma-2 expression in postmortem brain tissue from late-stage AD patients [33]: how sigma-2 expression is changed in AD remains further study.

Sigma-2 receptor antagonists can be used to treat AD. In 2014, the role of sigma-2 receptor in AD was first studied by Izzo. He identified several compounds that could reverse Aβ oligomer-induced synapse loss in neuronal culture, as well as the memory deficits in AD mouse models [124]. Later, these compounds were identified as sigma-2 receptor ligands. In his other paper, the sigma-2 receptor was shown to act as an Aβ oligomer receptor in neurons; its specific antagonists and antibodies prevented the binding of Aβ oligomer to synaptic puncta in vitro [33]. In addition, the expression of the sigma-2 receptor is upregulated upon the treatment of Aβ oligomers [33]. Consistent with this notion, sigma-2 receptor antagonist, SAS-0132, protects the *C. elegan* AD model from neurodegeneration. Further, it rescues memory deficits in the AD mouse model [28]. Another sigma-2 receptor antagonist, CT1812, also prevents and displaces the binding of Aβ oligomers to neurons and has entered clinical trial II to treat AD [32].

#### 4.3.2. The Potential Mechanism of Sigma-2 Receptor Ligands in AD

Although multiple mechanisms have been proposed to explain the anti-AD effects of sigma-2 ligands including its neuroprotection and anti-inflammation. They may work via its effects on cholesterol homeostasis to modulate lipoprotein trafficking, Aβ production, tau hyperphosphorylation, and neuroinflammation.

The disruption of lipoprotein trafficking may contribute to the actions of sigma-2 ligands in AD. In the brain, ApoE binds cholesterol to form lipoprotein particles, then enters into neurons by LDLR and LRP1 [51]. ApoE is also associated with Aβ42. Recently, it has been shown that the inhibition of TMEM97/sigma-2 reduces Aβ42 and ApoE uptake in primary rat cortical neurons [27]. Increased uptake of Aβ42 results in accumulation and aggregation within neurons eventually leading the formation of plaques and neuronal death [125], which contributes to AD pathogenesis. This study indicates that TMEM97/sigma-2–PGRMC1-LDLR trimeric complex might be a potential target to reduce Aβ42 accumulation in neurons [27].

Sigma-2 ligands might also modulate Aβ production via cholesterol. Cholesterol has been shown to directly modulate APP cleavage in neuronal cultures by promoting β- and γ-secretase activity [126]. Decreasing membrane cholesterol levels using β-methyl-cyclodextrin (βMCD) reduces Aβ generation via the inhibition of β secretase and γ-secretase [127]. In addition, using cell-free assay, it has been demonstrated that cholesterol could directly regulate the activities of recombinant β secretase and γ-secretase [128]. Cholesterol could affect the activities of secretases via its modulation on lipid rafts. The increase in membrane cholesterol promotes the association of APP, β-, and γ-secretases within the lipid rafts and increases Aβ production [129,130]. Similarly, sigma-2 ligands also affect tau phosphorylation via cholesterol. Cholesterol controls Aβ-induced tau proteolytic cleavage by calpain [131]. In addition, hyperphosphorylated tau is also present in lipid rafts, indicating cholesterol might have ability to modulate tau hyperphosphorylation [132].

Furthermore, sigma-2 ligands could modulate neuroinflammation via oxysterols in AD. LXRs activated by oxysterols inhibit inflammatory gene expression, since LXRs could bind and inactivate pro-inflammatory genes [133]. Moreover, LXR activation may prevent neuroinflammation by indirectly down-regulating TLR target genes [134]. However, LXR-activating oxysterols might promote inflammation independently of LXRs; 27-HC, 24-HC, and 7β-HC enhanced inflammatory gene expression in human neuroblastoma SH-SY5Y cells via TLR4/cyclooxygenase-2/prostaglandin E synthase [135].

## 5. Conclusions

More and more evidence has shown that the sigma-2 receptor might present a potential avenue for treating cancer and AD; the mechanisms connecting these two diseases are unknown. There is increasing proofs that they converge on a common pathological hub that involves cholesterol homeostasis. Data obtained over the past few years from human and animal models indicate that cholesterol metabolism is altered in AD and cancer, underscoring the importance of cholesterol homeostasis in AD and cancer. Future studies should aim to resolve the question of whether sigma-2 ligands could modulate cholesterol homeostasis to treat AD and cancer. In addition, the molecular and cellular mechanism of their potential benefits in AD and cancer also needs to be studied. Although the molecular identity of the sigma-2 receptor has been identified as TMEM97, the lack of its structural information has still severely hindered the understanding of its physiological roles, its signaling pathways, and the development of more selective sigma-2 ligands. In addition, sigma-2 ligands (usually its antagonists) are claimed to be neuroprotective in AD, while its agonists are reported to be cytotoxic in cancer. The basis for these opposing outcomes remains unknown.

## Figures and Tables

**Figure 1 molecules-25-05439-f001:**
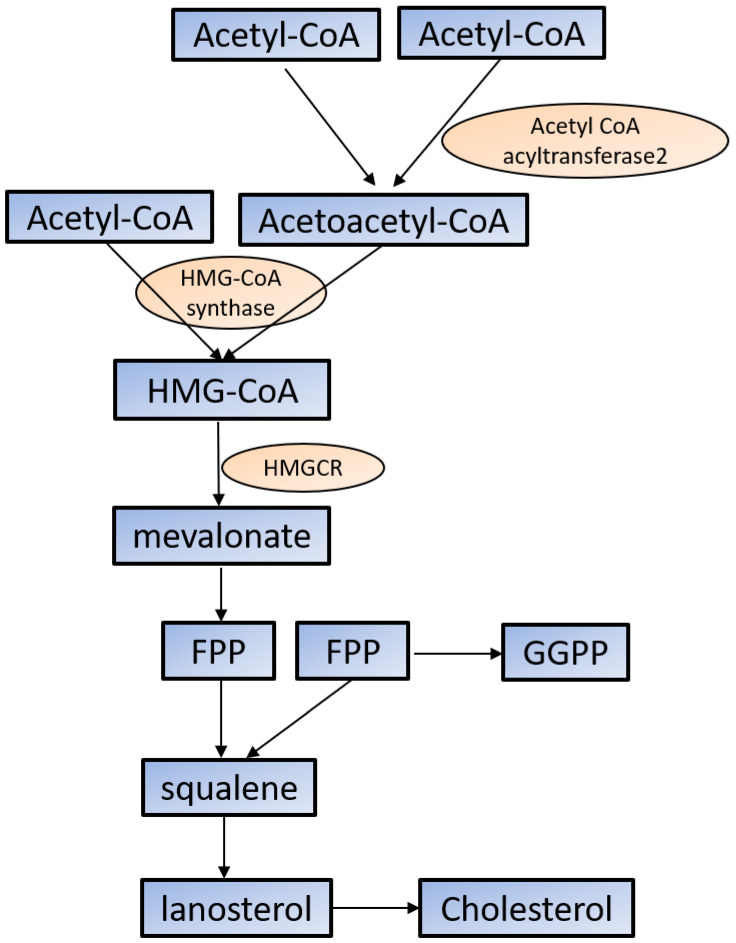
Cholesterol synthesis in the cytoplasm. Cholesterol biosynthesis begins with acetyl-CoA. See the text for the detailed information. HMG-CoA: 3-hydroxy-3-methylglutaryl-CoA; HMGCR: HMG-CoA reductase; FPP: farnesyl pyrophosphate; GGPP: geranylgeranyl pyrophosphate.

**Figure 2 molecules-25-05439-f002:**
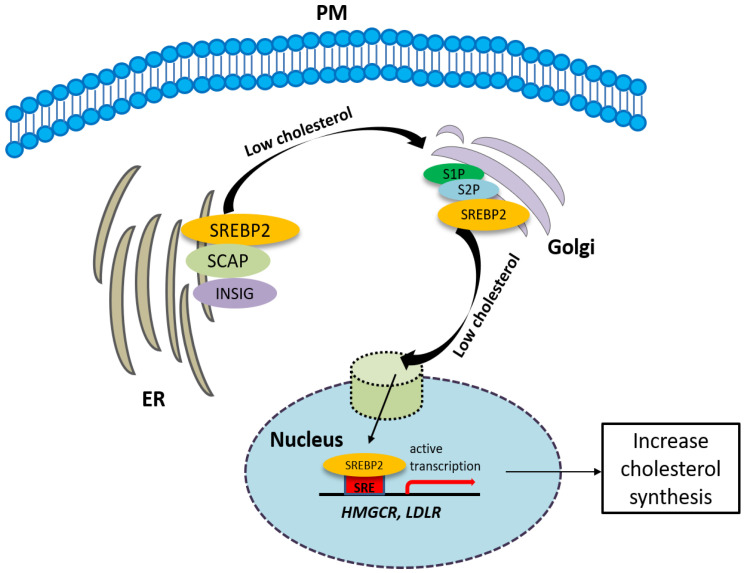
The synthesis of cholesterol is regulated by feedback control involving sterol regulatory element-binding protein 2 (SREBP2). See the text for the detailed information.

**Figure 3 molecules-25-05439-f003:**
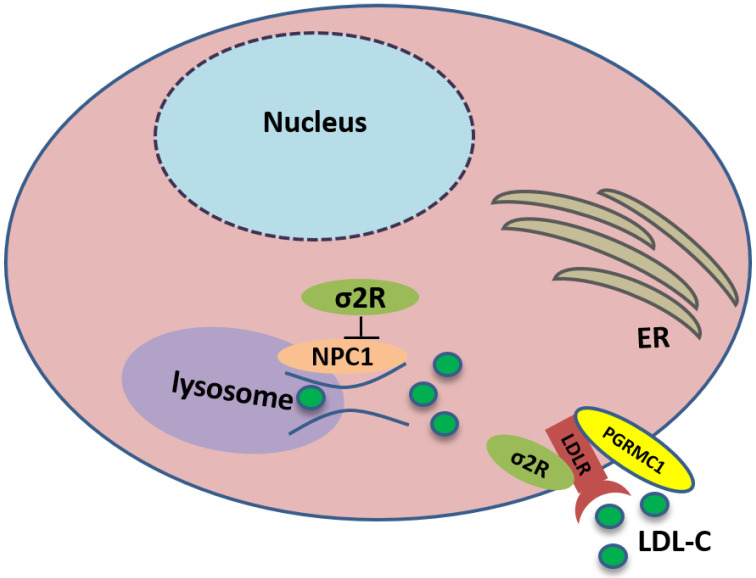
Sigma-2 receptor controls the trafficking of cholesterol. TMEM97 forms a trimeric complex with progesterone receptor membrane component 1 (PGRMC1) and low-density lipoprotein receptor (LDLR), which is responsible for the internalization of LDL. In addition, TMEM97 is a Niemann-Pick C1 (NPC1) binding protein and controls the trafficking of cholesterol out of lysosomes.

**Figure 4 molecules-25-05439-f004:**
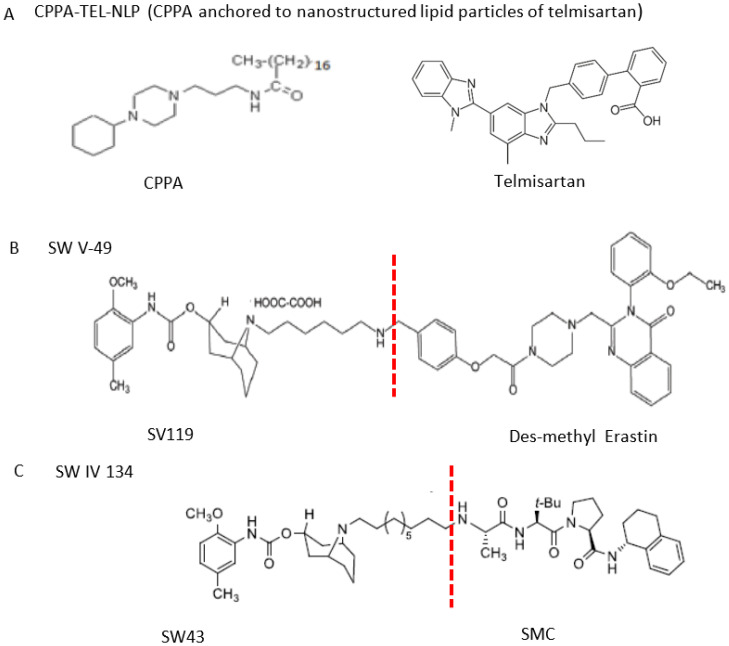
The structures of CPPA-TEL-NLP, SW V-49, and SW IV 134. (**A**) The structure of CPPA and Telmisartan: CPPA-TEL-NLP was prepared by anchoring CPPA to nanostructured lipid particles of Telmisartan. (**B**) The sigma-2 ligand SV119 was chemically conjugated to dm-Erastin, resulting in the sigma-2/dm-Erastin conjugate SW V-49. (**C**) SW IV 134 was constructed by sigma-2 ligand SW43 and small molecule SMAC mimetic compound (SMC). CPPA-TEL-NLP: 3-(4-cyclohexylpiperazine-1-yl) propyl amine (CPPA) anchored nanostructured lipid particles (NLP) of telmisartan (TEL).

**Figure 5 molecules-25-05439-f005:**
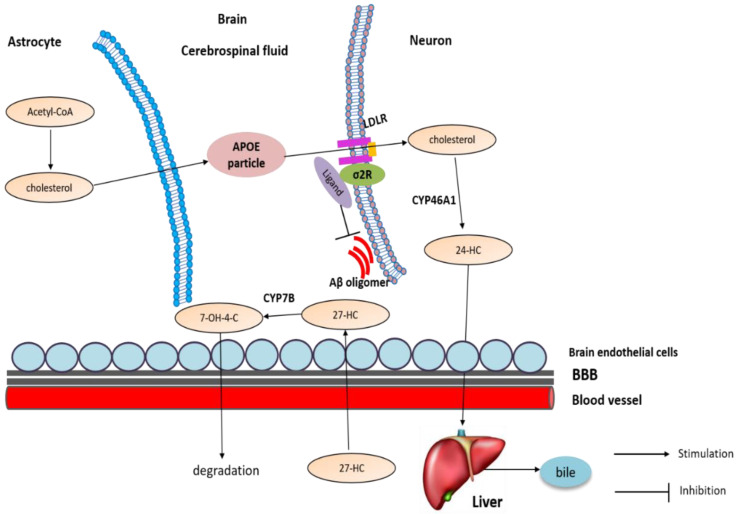
Cholesterol transport and metabolism in brain. Neurons in the adult do not efficiently synthesize cholesterol; they rely on the input from astrocytes as an external source. In neurons, CYP46A1 converts excessive cholesterol into 24-HC, then it is exported out of the brain and carried by LDL to the liver for degradation, while another oxysterol, 27-HC, is produced by CYP27A1 in the periphery and moves into the brain by circulation. Then, 27-HC is converted into 7−OH-4-C by the enzyme CYP7B, which then diffuses out of the brain through the BBB and moves into the periphery for degradation.

**Table 1 molecules-25-05439-t001:** The pharmacology of sigma-2 ligands.

Compound	Putative Action	Assays Used	Reference
CB-184 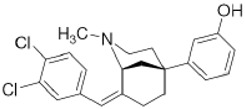	Agonist (sigma-2/TMEM97)	Apoptosis assay, Lactate dehydrogenase (LDH) release	[30,31]
CM398 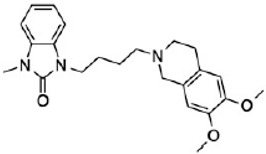	? (sigma-2/TMEM97)		[23]
CT1812 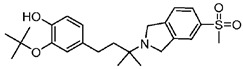	Antagonist (sigma-2/PGRMC1)	Trafficking assay	[32,33]
DKR-1677 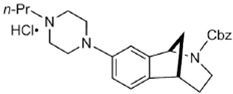	? (sigma-2/TMEM97)		[22]
DTG 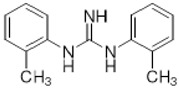	Agonist (sigma-1 and sigma-2/TMEM97)	Apoptosis assay, LDH release	[34,35,36]
JVW-1034 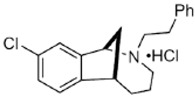	? (sigma-2/TMEM97)		[21]
PB221 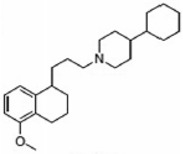	Agonist (sigma-2/TMEM97)	(3-(4,5-dimethylthiazol-2-yl)-2,5-diphenyltetrazolium bromide (MTT) assay	[19]
PB28 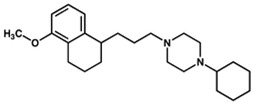	Agonist (sigma-2/TMEM97)	Apoptosis assay, in vivo tumor xenografts	[35,37]
RHM-4 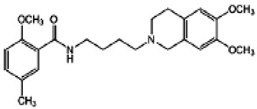	Antagonist (sigma-2)	3-(4,5-dimethylthiazol-2-yl)-5-(3-carboxymethoxyphenyl)-2-(4-sulfophenyl)-2H-tetrazolium (MTS) cell viability assay, Caspase-3 activation assay	[29]
SAS-0132 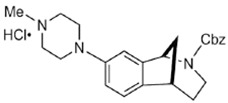	Antagonist (sigma-2/PGRMC1)	Ca^2+^ assay	[28]
Siramesine 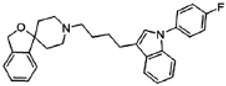	Agonist (sigma-2/TMEM97)	LDH release, apoptosis assay, in vivo tumor xenografts	[38,39,40]
SV119 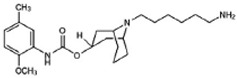	Agonist (sigma-2/TMEM97)	LDH release, apoptosis assay, in vivo tumor xenografts	[41,42,43]
UKH-1114 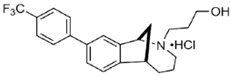	Agonist (sigma-2/TMEM97)	Pain relief	[20]
WC-26 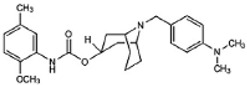	Agonist (sigma-2/TMEM97)	Apoptosis assay, LDH release	[42,43,44]

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
