# Peer review of "Sigma-2 Receptor—A Potential Target for Cancer/Alzheimer’s Disease Treatment via Its Regulation of Cholesterol Homeostasis"

_molecules, 2020, doi:10.3390/molecules25225439_

Round 1

Reviewer 1 Report

Alzheimer's disease (AD) and cancer are the cause of many deaths and are both medical and social problems. The results of the current research indicate the link between the molecular mechanisms involved in various types of cancers and AD. Moreover some anticancer agents have been used in the treatment of AD. The epidemiological data revealed an inverse correlation between the two diseases and some data shown that compared to people without Alzheimer’s dementia the risk of developing cancer is lower among patients with AD while the risk of developing AD is lower in patients with a history of cancer. The search for mutual correlations and mechanisms between these two diseases seems necessary.

Presented review describes sigma-2 receptor, its ligands and possible mechanisms regulation of cholesterol homeostasis, which in turn could be beneficial for the treatment of both cancer and AD.

Suggestions for a correction.

The structure of the compounds should be completed in Table 1, molecule symbols are insufficient, the chemical structure of the receptor ligands is important.

I suggest correction  the Part 3.3.3. Sigma-2 receptor ligands as anticancer drug delivery vehicles.

The description should be completed by adding a figure, e.g. by means of an additional drawing containing the linkage structures of the anti-cancer drugs (Telmisartan, des-methyl Erastin)    with ligands of the sigma-2 receptor (CPPA, SV119) respectively, as well as structures of SW43 and SW IV-134.

Author Response

1. The structure of the compounds should be completed in Table 1, molecule symbols are insufficient, the chemical structure of the receptor ligands is important.

A: the reviewer’s point is well accepted; we have added the chemical structure of the receptor ligands mentioned in the manuscript into Table 1 (page 3-5).

2. I suggest correction  the Part 3.3.3. Sigma-2 receptor ligands as anticancer drug delivery vehicles.

The description should be completed by adding a figure, e.g. by means of an additional drawing containing the linkage structures of the anti-cancer drugs (Telmisartan, des-methyl Erastin)    with ligands of the sigma-2 receptor (CPPA, SV119) respectively, as well as structures of SW43 and SW IV-134.

A: This point is well taken, we have added figure 4 (page 11) to demonstrate the structures of SW IV-134, CPPA-TEL-NLP and SW V-49 including their components.

Reviewer 2 Report

The authors provide a literature review of Sigma-2 receptor ligands' potential use in cancer and Alzheimer's disease in the paper. I want to congratulate the authors for this well-written review article with a nice flow of information. The authors' have provided a brief historical context about the identification and characterization of Sigma-2 receptors, and offered a current understanding of "trimeric-complex" involving Sigma-2 receptors along with PGRMC1 and LDLR. The paper goes on to detail the synthesis and transport of cholesterol and the putative role of Sigma-2 receptors in both cholesterol biosynthesis and trafficking. Citing several papers, the authors have compiled a complicated role of cholesterol in cancer (where it can be both pro-proliferative and pro-apoptotic) and AD. The paper reviews several studies leveraging Sigma-2 receptor expression in several cancers to use Sigma-2 agonists for imaging in cancer diagnosis and drug delivery systems for other small anti-cancer agents. Furthermore, it provides examples of studies which indicate direct anti-cancer activity of Sigma-2 agonists, and provides a current understanding of the possible underlying mechanism of such effects, including disruption of lipid rafts and modulation of the tumor microenvironment. Similarly, the paper provides examples of studies targeting the role of cholesterol homeostasis and, therefore, the putative role of Sigma-2 receptors in the treatment of AD. In this paper, the authors have compiled several studies that indicate a growing appreciation of Sigma-2 receptors for targeting both cancer and AD. Although several questions remain unanswered, including those indicated by the authors e.g., the differential role of agonists and antagonists in cancer and AD, seemingly low efficacy of nanomolar binding cytotoxic ligands, etc. This paper will provide an excellent primer for the researchers/students interested in the field, and therefore I recommend acceptance of this review paper after some minor corrections.

  1. Figure 2 and Figure 4, have some of the text outside the margin lines
  2. Please recheck English phrasing in certain sentences. e.g. line 43, 111, 361, etc. This is not exhaustive list but please do re-check. 

Author Response

1. Figure 2 and Figure 4, have some of the text outside the margin lines

A: We are sorry for this mistake and have adjusted figure 2 (page 7) and figure 4 (it is figure 5 now, see page 13), so all the text of these figures is within the margin lines.

2. Please recheck English phrasing in certain sentences. e.g. line 43, 111, 361, etc. This is not exhaustive list but please do re-check. 

A: Thanks for reviewer’s suggestion, we rephrased sentences in line 43, 94, 111, 281, 361.

Line 43 “ However, later studies discarded this hypothesis” was changed to line 43 “ However, this hypothesis was discarded”.

Line 94 “Consistent with this notion, the activation of TMEM97 by its ligands” was changed to line 117 “ Consistently the activation of TMEM97 by its ligands”.

Line 111 “ Cholesterol is necessary for not only membrane integrity and fluidity” was changed to line 160 “Cholesterol is required for not only membrane integrity and fluidity”.

Line 281 “ In addition, SW IV-134 is comprised of a sigma-2 receptor ligand (SW43)” was changed to line 348 “ In addition, SW IV-134 is constructed by the conjugation of a sigma-2 receptor ligand (SW43)”.

Line 361 “Using in vitro quantitative receptor autoradiography, it has demonstrated that both sigma-1 and sigma-2 receptors are widely distributed in the rat brain” was changed to line 470-471 “ Using in vitro quantitative receptor autoradiography technique, both sigma-1 and sigma-2 receptors are found to be widely distributed in the rat brain”